# Psychological Experiences of Parents of Pediatric Cancer Patients during and after COVID-19 Pandemic

**DOI:** 10.3390/cancers16050891

**Published:** 2024-02-22

**Authors:** Antonella Guido, Elisa Marconi, Laura Peruzzi, Nicola Dinapoli, Gianpiero Tamburrini, Giorgio Attinà, Alberto Romano, Palma Maurizi, Stefano Mastrangelo, Silvia Chiesa, Maria Antonietta Gambacorta, Antonio Ruggiero, Daniela Pia Rosaria Chieffo

**Affiliations:** 1Clinical Psychology Unit, Fondazione Policlinico Universitario A. Gemelli IRCCS, 00128 Rome, Italy; guido.antonella@gmail.com (A.G.); elisa.marconi@policlinicogemelli.it (E.M.); laura.peruzzi@guest.policlinicogemelli.it (L.P.); danielapiarosaria.chieffo@policlinicogemelli.it (D.P.R.C.); 2Pediatric Oncology Unit, Fondazione Policlinico Universitario A. Gemelli IRCCS, 00128 Rome, Italy; giorgio.attina@policlinicogemelli.it (G.A.); albertoromano90.ar@gmail.com (A.R.); palma.maurizi@unicatt.it (P.M.); stefano.mastrangelo@unicatt.it (S.M.); 3Radiation Oncology Department, Fondazione Policlinico Universitario A. Gemelli IRCCS, Università Cattolica Sacro Cuore, 00128 Rome, Italy; nicola.dinapoli@policlinicogemelli.it (N.D.); silvia.chiesa@policlinicogemelli.it (S.C.); mariaantonietta.gambacorta@unicatt.it (M.A.G.); 4Pediatric Neurosurgery, Fondazione Policlinico Universitario A. Gemelli IRCCS, Università Cattolica Sacro Cuore, 00128 Rome, Italy; gianpiero.tamburrini@unicatt.it; 5Department of Woman and Child Health and Public Health, Università Cattolica Sacro Cuore, 00168 Rome, Italy; 6Department of Life Sciences and Public Health Department, Università Cattolica Sacro Cuore, 00168 Rome, Italy

**Keywords:** stress, quality of life, parent perception, COVID-19, cancer, children

## Abstract

**Simple Summary:**

This study aimed to evaluate how the COVID-19 pandemic affected the psychological well-being of parents of pediatric oncology patients two years after the pandemic started. The authors’ objective was to make a detailed comparison between the data collected in the current study and the data gathered in the previous research with the aim of observing any changes, whether for better or worse, at the psychological level in the caregivers of the patients in question. This research could be of great use for monitoring the psychological health of the sample and for being able to intervene promptly in the event of the worsening of the reported symptoms.

**Abstract:**

Background: Family members dealing with the devastating impact of a cancer diagnosis are now facing even greater vulnerability due to the COVID-19 pandemic. Alongside the already overwhelming trauma, they must also bear the distressing burden of the infection risks. The purpose of this study was to examine and explore the effects in parents of pediatric cancer patients two years after the start of the COVID-19 pandemic to compare these data with the previous data. Methods: We conducted a single-center prospective observational study, enrolling 75 parents of 42 pediatric oncology patients. Four questionnaires (IES-R; PSS; STAI-Y and PedsQL) were given to the parents 2 years after the first evaluation. Results: The bivariate matrix of correlation found a strong significant positive correlation between IES-R and PSS scores (r = 0.526, *p* < 0.001) as in T1. Stress symptoms (t = 0.00, *p* < 0.001) and levels of anxiety (trait) (t = 0.32, *p* < 0.001) remained unchanged; anxiety state levels appeared to have increased (t = 0.425, *p* < 0.001); there was a significant decrease in the PedsQL tot (t = 5.25, *p* < 0.001). Conclusions: The COVID-19 pandemic has influenced the levels of stress and anxiety of parents and the quality of life of patients, also correlating with the traumatic impact of the diagnosis.

## 1. Introduction

In 2020, the world’s population faced a health emergency related to COVID-19, which has been described as a global health crisis [1] and which has decreed changes that are still lasting today [2]. The uniqueness of this event was also characterized by the containment measures adopted worldwide to combat the spread of the disease. For example, in Italy, the first country in Europe to adopt the national blockade, the population experienced a quarantine that lasted from 9 March 2020 to 18 May 2020 [3,4]. 

In the current health emergency, an increase in emotional and behavioral symptoms in children has been reported along with regressive behaviors predicted by the change in sleep quality, boredom and psychological difficulties of mothers [5]. An increased risk to the pediatric population with pre-existing mental and physical disabilities has also been reported [6,7,8].

The COVID-19 pandemic has had an important psychological impact on the general population’s mental health [9], both adult and pediatric [5,9]. In general, families have reported negative effects of COVID-19 [10], and the literature documents consequences for the mental health of adults in terms of post-traumatic stress disorder and also acute stress disorder and attachment disorder [11]. 

Among the populations most affected by the restrictive measures were cancer patients [12,13], especially those of pediatric age, and their families. The psychological consequences of the pandemic, and in particular the post-pandemic phase, on this population are still unclear. Some research points out that children with cancer and their parents did not report worse psychopathological outcomes following the pandemic [14,15]. Other studies, instead, have shown an increased emotional and psychological burden in parents and also in children [16,17]. It might be useful to investigate the traumatic aspects and sequelae in this specific population [16].

The spread of the COVID-19 infection has created alarm in the population of cancer patients about the clinical vulnerability to which they are exposed [18] and also in caregivers. The emotional experience related to the sudden emergency, of alertness or sense of isolation, may be aggravated by the condition related to the oncological disease. 

Pediatric cancer is in fact a traumatic risk condition for patients [19,20] and parents [21]. Numerous researchers emphasize the profound psychological distress experienced by parents of children diagnosed with cancer, as evident from their depressive symptoms [22] and post-traumatic stress symptoms [23,24,25]. Overall, the well-being of these parents appears to be significantly lower than that of parents with healthy children [26]. In 2020, the study authors examined how the COVID-19 pandemic affected parents of children with cancer. We specifically aimed to examine how this heightened the feelings of uncertainty and fragility among family members who, alongside the emotional impact of the cancer diagnosis, grappled with the concerns tied to the risk of infection.

In our study, started during the first lockdown, parents of pediatric oncology patients showed clinically significant anxiety levels, stress levels and a high risk for post-traumatic symptoms [17].

Other research shows that parents of children with cancer reported an increased fear of viral infections and a desire to isolate the sick child from everyone except the caregiver [16]. In addition, Darlington et al. 2021 reported anxiety, fears and concerns about the psychological, social and economic impact of isolation in the parents of these children [16].

An Australian study shows, despite the low rate of cases, that the COVID-19 event negatively affected family well-being [27]. 

As noted, these results were not confirmed by other studies, which showed that few caregivers observed distress at the beginning of the pandemic compared to the pre-COVID-19 period [15].

Considering the presence in the literature of contrasting data on the effects of COVID-19 and the extreme vulnerability of the oncological population, we monitored the psychological state of the same parents evaluated in the first study [17] over time to see how it changed and compared the data after 2 years. Our group analyzed, during the first lockdown (T1) and subsequently after 2 years (T2), both the psychological impact on parents of their children’s cancer diagnosis, and the impact related to the epidemic event and their correlation. From the first observation, we reassessed their psychological status due to the distress caused by the pandemic. 

The objective was to assess how the COVID-19 pandemic had affected the psychological state of parents of children with cancer and specifically the levels of stress, anxiety, and perceived quality of life.

## 2. Materials and Methods

We conducted a single-center prospective observational study, enrolling parents of pediatric oncology patients. In the first phase of the study (T1), parents were enrolled in the acute phase of the pandemic for a three-month period from June to August 2020 [17]. In this study, we enrolled the same sample of parents two years later (T2) to compare the data with the initial study. The parents were divided into two groups: those with children in treatment (GT) and those with children who had completed treatment (GOT).

The sample was selected considering data from the literature, reporting very high anxiety levels in parents of pediatric cancer patients after their child’s diagnosis, followed by a reduction in the first 3 months of up to 66% [28,29]. The present study aims to investigate whether two years after the COVID-19 event (T2), the levels of anxiety, traumatic impact and stress in parents have changed compared to what was detected in T1.

The primary objective of the initial study was to determine the correlation between the Impact of Event Scale—Revised and the Perceived Stress Scale, with a significance level of 0.05 (two-tailed) for rejecting the null hypothesis and a power of 0.80 (type II error rate) to fail to reject the null hypothesis based on the alternative hypothesis. To achieve this, a sample size of 85 cases [30] was analyzed. However, due to the conclusion of the recruitment period, the desired number of participants could not be fully attained.

### 2.1. Participants

The study was offered to all parents during the COVID-19 pandemic phase June–August 2020 and two years later. The study from the outset included monitoring the psychological state of the parents over time. For this reason, further measurements were carried out after the first survey on the same population.

In our study, at T1, 80 parents of children with solid and hematological tumors were enrolled; the study was carried out at the Pediatric Oncology Unit, the Pediatric Neurosurgery Unit and the Radiotherapy Unit of the Fondazione Policlinico Universitario A. Gemelli IRCCS in Rome.

At T2, the number of parents re-evaluated was 74 (Table 1) for a total of 42 patients (Table 2) involved. Compared to T1 (*n* = 45), two patients had died and one parent had returned to his country of origin.

At the first assessment (T1) the following questionnaires were distributed: Impact of Event Scale—Revised (IES-R); Stress Scale (PSS), State–Trait Anxiety Inventory (STAI-Y), Pediatric Quality of Life Inventory Parent Proxy report (PedsQL). At T2 (two years later), the 4 selected questionnaires (IES-R, PSS, STAY and PedsQL) were distributed once again. We analyzed the preliminary data from this sample and the evolution of the scores over time.

The selection criteria for the subjects in this study were carefully chosen to ensure the inclusion of parents who met specific conditions. Firstly, the subjects had to be parents of patients diagnosed with cancer and enrolled at T1. Secondly, they had to be parents of patients who were either currently undergoing treatment or had completed their treatment regimen at least five years ago. Lastly, the subjects had to be parents of patients who were under the age of 25. The patients who were under the age of 25 and recruited for this study were those who were currently receiving treatment for pediatric cancer or were in the follow-up stage. However, parents with psychiatric or cognitive disorders, as well as parents with intellectual disabilities, were excluded from the study. To ensure the accuracy of the results, the parents who were recruited for the study underwent screening by the psychology service.

Parents who had been diagnosed with psychiatric disorders were also excluded from the study. This was carried out to maintain the integrity of the research and ensure reliable data.

Furthermore, this study was conducted in accordance with the Helsinki Declaration, which sets ethical guidelines for medical research involving human subjects. Additionally, the study was approved by the Institutional Review Board to ensure the protection and well-being of all participants. Prior to participating in the study, written informed consent was obtained from all parents involved, further ensuring their understanding and willingness to participate.

### 2.2. Measures

#### 2.2.1. Impact of Event Scale—Revised (IES-R)

The Impact of Event Scale—Revised (IES-R) is a tool used to assess the emotional response to a specific traumatic event in adults. This comprehensive 22-item self-report scale measures subjective stress through various components. These include the total subjective stress scale as well as three impactful subscales: avoidance, intrusion, and hyperarousal. By utilizing a 5-point rating system, ranging from “not at all” to “extremely”, individuals can effectively communicate the intensity of their experiences. To determine a high risk of PTSD symptomatology, a threshold of 33 has been established based on extensive research and literature analysis [31,32].

#### 2.2.2. Perceived Stress Scale (PSS)

The Perceived Stress Scale (PSS) is a psychological tool designed to assess one’s perception of stress. By probing into one’s feelings and thoughts over the past few months, the PSS provides insights into stress levels [10,33]. With a concise format of 10 items, the PSS-10 offers a self-report questionnaire that individuals can readily complete. Using a 5-point Likert scale, where 0 signifies “never” and 4 represents “very often”, every item contributes to the overall assessment. It is worth noting that the PSS-10 comprises six positively worded items (1, 2, 3, 6, 9, and 10) forming the positive factor, along with four negatively phrased items (4, 5, 7, and 8) encompassing the negative factor. Scores on this scale range from 0 to 40, with higher scores indicating a greater perception of stress. To further understand the significance of the scores, they can be categorized into three bands: low stress (0–13), moderate stress (14–26), and high perceived stress (27–40). During the challenging times of the pandemic, this scale emerged as an useful tool not only in Italy but also in various other countries [34,35]. Its reliability, as indicated by Cronbach’s alpha, has been reported to range from 0.67 to 0.91, reflecting internal consistency.

#### 2.2.3. Spielberger State–Trait Anxiety Inventory (STAI-Y)

The Spielberger State–Trait Anxiety Inventory (STAI-Y) is a comprehensive 40-item questionnaire designed to evaluate both state anxiety (STAI-Y1) and trait anxiety (STAI-Y2) separately. With 20 items dedicated to each scale, this assessment provides an understanding of an individual’s temporary state of anxiety as influenced by their current situation as well as their general propensity to experience anxiety [36,37]. To ensure accurate results, scores exceeding 40 were considered significant, as this threshold minimizes the likelihood of false positive and negative outcomes [38,39]. Additionally, the internal consistency reliability of the STAI-Y is high, ranging from 0.91 to 0.95 for the state scale and from 0.85 to 0.90 for the trait scale.

#### 2.2.4. Pediatric Quality of Life Inventory TM (PedsQL 4)

The Pediatric Quality of Life Inventory TM (PedsQL) 4.0 Generic Core Scale is a tool for assessing the health-related quality of life of children. This parent proxy report covers various aspects including physical, emotional, social, and school functioning scales. The higher the scores on these scales, the better the quality of life experienced by the child. When it comes to pediatric oncology populations, the PedsQL has been extensively used and well validated. Numerous studies have demonstrated its reliability and validity, with Cronbach’s alphas consistently meeting or exceeding 0.70 [40,41]. Moreover, in pediatric cancer samples, the scale has also shown good construct validity.

Notably, the PedsQL showcases internal consistency reliability, particularly for the total scale score, with an alpha value of 0.90 in parent reports. Overall, the PedsQL is a reliable and robust instrument for evaluating the quality of life in pediatric populations.

### 2.3. Procedure

After a two-year period (T2), the parents received questionnaires in regard to the first study (T1). The researchers took the time to explain the intention behind the study to the parents. Additionally, prior to the commencement of the study, written consent was obtained from the parents, ensuring their willingness to participate. The parents were also assured that all the information they provided would be kept strictly confidential. At the time of the second assessment, it was explained to the parents that they could withdraw from the study at any time, and they were reminded that this would have no impact on the care provided to them or their child, as clearly stated in the signed consent. Psychologists from the Clinical Psychology Unit of Fondazione Policlinico Universitario Agostino Gemelli IRCCS conducted the assessments just like in T1. It was noted that a majority of parents opted for interviews rather than completing the questionnaires on their own. Each patient participating in the study had their questionnaires filled out by their respective parents individually. The parents were explicitly informed that the IES-R scale focused on their child’s cancer diagnosis, while the remaining questionnaires pertained to the ongoing phase of the pandemic.

### 2.4. Statistical Analysis

We performed descriptive statistics to assess the sample age, sex, cancer diagnosis and treatment status. For the main study variables, the observed mean and standard deviation were calculated. We subsequently compared the results of our sample with the cut-offs of the administered questionnaires. We created a correlation matrix to examine the relationships between the scores of the four scales. This matrix allowed us to see the correlations between the questionnaires (IES-R, PSS, STAI-Y, PedsQL) at two different time points. *p*-values were two-tailed; statistical significance was set at *p* < 0.05. After that, we conducted comparisons between the treatment and off-therapy groups as well as between mothers and fathers. To ensure accuracy, we used the Mann–Whitney U test, a reliable statistical method for analyzing nonparametric samples. Finally, comparisons were made between the trend of scores in the course of the two T1 and T2 measurements. Statistical analysis was performed using R version 4.0.3.

## 3. Results

At the time of the second assessment, 75 parents out of the initial 80 agreed to participate. The study examined the parents of 42 patients, consisting of 29 individuals with solid tumors and 13 with malignant hematological diseases. These parents were categorized into two groups: those who had finished their treatment (off-therapy group, *n* = 26) and those who were currently undergoing treatment (in-treatment group, *n* = 16). Two years after the first study, the enrolled parents were numerically lower because one parent had returned to his country of origin and therefore could not be reached. Two children had died, so four parents were not enrolled in T2. A total of 75 parents, including 40 mothers and 35 fathers, participated in the study. All parents have provided written informed consent and agreed to participate. The demographic characteristics of the participants are outlined in Table 1. The sample showed significantly high levels of traumatic disorder risk (IES-R, $\bar{x}$ = 39.6 ± 16.35) and a prevalent presence of stress symptoms (PSS, $\bar{x}$ = 18.95 ± 5.17) compared to the average of the Italian population. Additionally, the parents in our sample displayed notable levels of anxiety, surpassing the STAI-Y cut-off scores. Specifically, the average values were Y1 (state), $\bar{x}$ = 42.95 ± 6.58 and Y2 (trait), $\bar{x}$ = 40.95 ± 4.78.

Correlation matrix. The bivariate correlation matrix revealed compelling findings. A strong, significant positive correlation was observed between IES-R and PSS scores (r = 0.526, *p* < 0.001), consistent with the findings from two years ago. Additionally, a positive correlation was found between PSS and STAI-Y2 (trait) scores (r = 0.305, *p* < 0.001), showing an even more significant association than in the previous assessment. Moreover, a correlation was identified between PSS and the PedsQL (emotional needs) scale (r = 0.344, *p* < 0.001), mirroring the findings from two years earlier.

*Groups comparison.* The division of groups into off- and on-therapy clearly demonstrated a significant impact of this variable on the outcome at T1 on IES-R scores (*p* < 0.001; off-therapy, $\bar{x}$ = 37.8 ± 16.9; on-therapy, $\bar{x}$ = 43.9 ± 16.1). However, these differences were not observed at T2. Furthermore, meaningful distinctions were found in subsequent comparisons between the scores of mothers and fathers on the PSS (*p* = 0.03; $\bar{x}$ = 20.3 ± 5.3; $\bar{x}$ = 17.6 ± 5.94) and IES-R (*p* = 0.04; $\bar{x}$ = 43.5 ± 17.1; $\bar{x}$ = 35.7 ± 16.6) scales, in which mothers had higher scores than fathers (Figure 1A,B); on the STAI-Y2 fathers had higher scores than mothers ($\bar{x}$ = 41.5 ± 5.06; $\bar{x}$ = 44.4 ± 5.5), but the difference did not reach levels of significance.

Time comparison (T1, T2). From the comparison of the study two years earlier and the present, it would appear that the presence of stress symptoms (t = 0.00, *p* < 0.001) and the levels of anxiety (trait) (t = 0.32, *p* < 0.001) remain unchanged, but the anxiety level state appears to have increased (t = 0.425, *p* < 0.001) from two years before. In general, in T2, there was a significant decrease in the PedsQL total (t = 5.25, *p* < 0.001), PedsQL Health (t = 3.9, *p* < 0.001), PedsQL Emotion (t = 3.09, *p* < 0.001) and PedsQL School (t = 4.17, *p* < 0.001). Lastly, the scores of the IES-R questionnaire in T2 showed a decrease in the IES-R scale (t = 1.084, *p* < 0.001).

## 4. Discussion

Pediatric cancer is a traumatic event for patients [19,20] and their parents [21], and in general, the well-being of these parents seems to be lower than that of parents of healthy children [22,26]. For this reason, cancer patients, especially pediatric patients, and their families may have been more exposed to psychopathological symptoms during the pandemic and post-pandemic phases, but the psychological outcomes have not yet been fully investigated. Some research has shown no worsening of psychological symptomatology in children with cancer and their parents during the pandemic compared with the past [14,15]. Other studies, on the other hand, have shown an increased emotional and psychological burden in parents and children [16,17]. The present study aimed to explore psychological symptoms in parents of pediatric cancer patients. In particular, we investigated the level of trauma impact, anxiety, and stress during and after the COVID-19 pandemic. Starting from the previous study [17] on the psychological impact of parents with children with cancer during COVID-19, the same sample was compared 2 years after the first evaluation. Our sample experienced high levels of traumatic impact [17]. Other studies confirm this evidence and showed significant negative psychological impact in parents, due to the level of emotional distress and changes in daily family life [42]. Other studies also documented the prevalence of post-traumatic stress symptoms at severe levels (21–44%) [43] and incidences of current cancer-related PTSD in parents (6.2–25%) [44] which is higher than among parents in the general population [45]. Other studies confirm that these parents report trauma-related symptomatology [46,47,48,49] of comparable severity to other individuals diagnosed with post-traumatic stress disorder (PTSD). In contrast, other studies have not supported a relationship between cancer-related PTSD and PTSS in parents of children [23,50,51]. Another study confirmed that a childhood cancer diagnosis can have an impact on the onset of depressive symptoms and anxiety in caregivers. Furthermore, the same study reports that caregivers’ levels of anxiety and depression are strongly associated with the age of the child [52].

### 4.1. Traumatic Impact

For this reason, it appears necessary to continue to investigate these issues. Our research confirmed significantly elevated levels of trauma disorder risk for cancer diagnosis. Furthermore, it was observed how stress related to the “cancer diagnosis” event of one’s child is currently lower than in the study carried out previously (T1) during the COVID-19 pandemic: it would appear that parents considered their child’s illness more traumatic during the lockdown period, while in the post-pandemic period, the traumatic impact of the illness appears lower, or this fact could be associated with elaboration processes which, over time, have facilitated the reduction in the impact of the traumatic event. These results confirm the vulnerability of this population which is more exposed to other stressful events which, after the children’s cancer diagnosis, can happen and the importance of monitoring traumatic risk levels over time.

### 4.2. Stress and Anxiety Levels

As regards the COVID-19 event, this sample showed the persistence over time (a comparison between T1 and T2) of a high presence of stress and high levels of anxiety with scores which reached and exceeded the STAI-Y cut off. The alarm triggered by the pandemic would appear to continue to have an impact on parents’ anxiety and stress levels.

For some time, research in the field of psychology has shown how the psychological difficulties of patients and their parents change considerably over time: it would seem that anxiety and stress reach their peak in the period immediately following the diagnosis and then return to generally normal levels after the first year of illness [53]. Specifically, a longitudinal study highlighted higher levels of anxiety, compared to the control group, in the very early stages of the disease, while after one or two years, the levels relating to these constructs did not show significant differences compared to the sample.

It could be useful to observe, with further studies, the long-term consequences caused by the pandemic and investigate the impact of the oncological disease.

### 4.3. Quality of Life

The analysis carried out also highlighted a significant reduction over time in the perceived quality of life. This data would seem to be an important outcome of the pandemic, which has negatively changed the quality of life of patients, according to parents.

In agreement with other studies, the lockdown period was experienced as a real collective trauma. The pandemic emergency has had a severe impact on the quality of life of the general population, as also confirmed by ISTAT data [54]. People felt a sense of confusion and a fear for their physical safety [55]. However, let us not forget that a decrease in quality of life could be the effect of the disease itself: indeed, some studies, in the field of pediatric oncology, show how high levels of anxiety and stress are associated with a compromise in social relationships and dysfunctional coping strategies, factors that lead to a decrease in the quality of life of the patient and his caregivers [56]. Whatever the cause of the decrease in quality of life, this variable needs to be monitored since a good current and future quality of life, to date, has become, together with recovery, the main aim of pediatric cancer treatment [57,58]. Our results also showed a positive correlation between parents’ distress and the child’s quality of life, as in the study two years earlier, and the result is consistent with the literature on the topic. Meta-analysis on a large sample of parents highlights a significant correlation evaluating the relationship between the quality of the child’s emotional and social life in relation to parental distress (r = 0.24, *p* < 0.001) [59].

The result of our study confirms how quality of life is a variable to be monitored during pediatric cancer treatment and how much it was impacted by the pandemic, which was a further traumatic event.

Our research found no statistically significant differences among parents of children with pediatric cancer. However, a meta-analytic review [60] showed an increase in mothers’ emotional distress, particularly at the time of diagnosis and initial treatment. One study demonstrated that gender differences can change over time following diagnosis [23]. As a result, the inclusion of parents at different disease stages, with different demographics and disease characteristics, contribute to mixed results that are not fully representative of parents’ experiences [61].

The present study is limited by the absence of a comparison group of parents who have not experienced the pandemic, and there is no control group of parents whose children do not have a cancer diagnosis. Further studies that can investigate and confirm the highlighted results will be important.

## 5. Conclusions

Our analysis confirmed the vulnerability of the pediatric cancer population which is more exposed to whatever other stressful events that occur, weighing on a delicate balance already burdened by the disease. The COVID-19 pandemic seems to have influenced the levels of stress and anxiety of parents and the quality of life of patients, also correlating with the traumatic impact of the diagnosis. In particular, the study highlights how much the parent’s psychological functioning influences that of the child, moving from a potential protective factor to a possible risk factor.

For these reasons it will be necessary to deepen these data with further research that can study and monitor the vulnerability of this population.

The study of levels of traumatic risk and vulnerability, over time, before other stressful events can highlight the need for preventive interventions on families who are experiencing and have experienced pediatric cancer.

Overall, as a recent review states [43], social and psychological support during the diagnosis and treatment of pediatric cancer can help parents in the long term. For these reasons, from a bio-psycho-social perspective, support services should be available during treatment but also afterwards, during off-therapy.

## Figures and Tables

**Figure 1 cancers-16-00891-f001:**
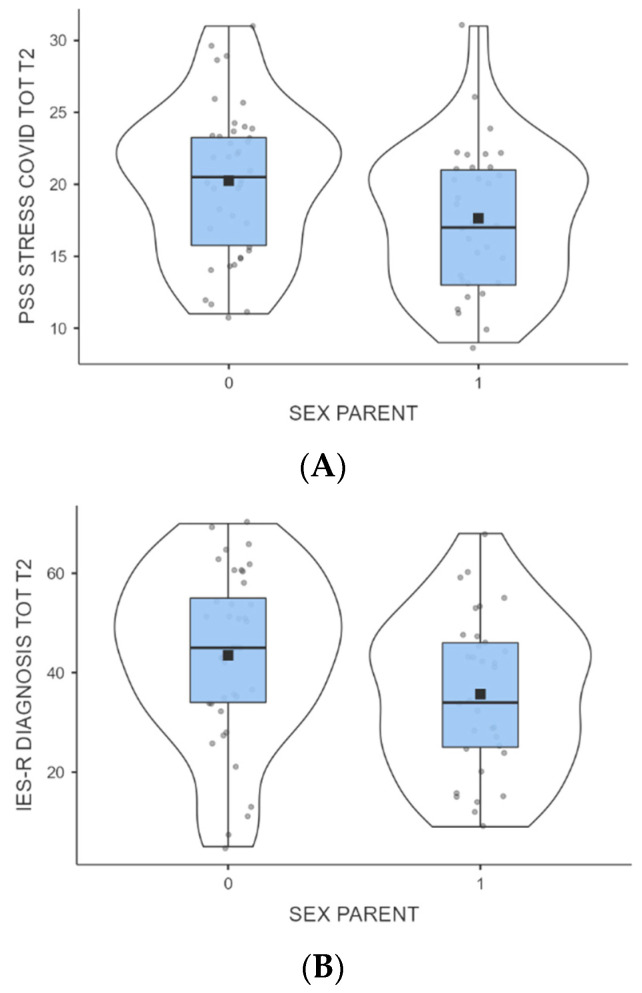
(**A**) Mothers and fathers PSS score at T2. (**B**) Mothers’ and fathers’ IES-R scores at T2.

**Table 1 cancers-16-00891-t001:** Characteristics of the population enrolled.

Parents (*n*)	75
Age at time of study (years)	49.34 ± 6.57
Mean range	32–52 years
**Relationship to patient**	
Mother	40
Father	35
**Level of parents schooling**	
Secondary school	28
High school and bachelor degree	33
Higher education	14
**Employment status**	
Housewife	17
Teacher	5
Employee	14
Nurse	5
Freelance	10
Laborer	10
Military employee	4
Artisan	9
Unemployed	1
**Numbers of children**	
Only child	8
More than one child	34

**Table 2 cancers-16-00891-t002:** Patients characteristics of enrolled parents.

Patients (*n*)	42
Age at diagnosis (years)	7.96 ± 5.62
Mean range	2–21 years
Age at time of study (years)	15.31 ± 6.86
Mean range	5–25 years
**Gender**	
Female	17
Male	25
**Cancer diagnosis**	
Leukemia	9
Lymphoma	4
Solid tumors	29
**Treatment status**	
In treatment	16
Off therapy (≥5 years)	26

## Data Availability

Data are contained within the article.

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
