# Peer review of "Psychological Experiences of Parents of Pediatric Cancer Patients during and after COVID-19 Pandemic"

_cancers, 2024, doi:10.3390/cancers16050891_

Round 1

Reviewer 1 Report

Comments and Suggestions for Authors

Title and Abstract:

1.        The title is appropriately descriptive of the study's content. The abstract provides a succinct overview of the study, methods, results, and conclusions, which is good practice. However, for an even more rounded abstract, consider including a sentence about the study's methodology and design to inform the reader right away about the type of research conducted.

Introduction:

2.        The introduction lays a solid foundation for understanding the context of the research. It could be enhanced by creating a more direct narrative that leads to the study's aims. The section presents the pandemic's impact broadly before focusing on the specific population of interest, which helps set the stage. However, ensuring a clear and direct path from the general problem to the specific research question would strengthen the section.

Materials and Methods:

3.        This section is detailed and provides a clear description of the study design, participant selection, and measures used. It would benefit from a clearer articulation of the research hypothesis or objectives, which seem to be implied rather than explicitly stated. The methods for data analysis are mentioned, but more detail on the statistical methods could be provided for the sake of transparency and reproducibility.

Results:

4.        The results are presented clearly with accompanying statistical data, which is good for validation of the findings. Nonetheless, this section could be improved by a more structured presentation, perhaps with subheadings for each main finding. The inclusion of graphs is a strong point, providing a visual representation of the data. However, the interpretation of these graphs within the text could be more thorough to guide the reader through the findings.

Discussion:

5.        The discussion is thoughtful, linking the findings back to the existing literature and the study's context. However, it could delve deeper into the implications of the findings, discussing potential reasons for the observed outcomes and considering broader implications. The limitations of the study are not explicitly stated, which is a critical component of any research discussion.

Conclusions:

6.        The conclusions effectively summarize the main findings and their significance. However, they could be expanded to include specific suggestions for future research or practical applications in clinical settings based on the study results. The section should ideally provide a clear take-home message for readers.

Tables and Figures:

7.        Ensure all tables and figures are clearly labeled and mentioned in the text.

Table 1 could be made more readable by aligning the numbers to the right, as it is standard for numerical data.

Overall Manuscript:

8.        Research Question and Purpose:

The research question and purpose of the study are clearly stated, aiming to investigate the psychological impact of the COVID-19 pandemic on parents of pediatric cancer patients. This includes examining levels of stress, anxiety, and perceptions of the child’s quality of life during the pandemic.

9.        Internal and External Logic:

The study's logic, both internally and externally, appears rigorous and clear. The methodology is well-defined, and the study's objectives are directly linked to the research question, providing a coherent framework for understanding the psychological impacts of the pandemic on this specific population.

10.     Literature Review:

The literature review adequately covers the relevant field, discussing the psychological impacts of the COVID-19 pandemic broadly and within the context of pediatric cancer patients and their families. It references recent studies and established theories, showing a good understanding of the existing research landscape.

11.     Language Structure and Coherence:

The language structure of the article is logically coherent, with a clear progression from introduction to methodology, results, and discussion. The argument is well-structured, with each section building upon the previous to support the study's findings.

12.     Argument Strength and Citation:

The argument is strong, supported by both the collected data and a wide range of citations from relevant literature. The study's findings are well-integrated into the broader context of psychological research related to health crises, pediatric cancer, and the effects of the COVID-19 pandemic.

13.     Academic Style and Language Consistency:

The language is consistent with the style of an academic paper, employing technical terms and concepts appropriate for a scholarly audience. The writing is formal, precise, and adheres to the conventions of academic writing in psychology.

Specific Sentences for Revision:

14.     "The COVID-19 pandemic is a pandemic event..." (Introduction) - This is redundant. Consider rephrasing to "The COVID-19 pandemic has been a significant global health crisis..."

15.     "The study was offered to all parents who attended the services described below during the COVID-19 pandemic phase June–August 2020." (Materials and Methods, 2.1 Participants) - It's unclear what "services described below" refers to. Specify these services immediately before or after this sentence for clarity.

Comments on the Quality of English Language

Need improvement.

Reviewer 2 Report

Comments and Suggestions for Authors

Psychological experiences of parents of pediatric cancer patients during and after COVID-19 pandemic

 Abstract

The Abstract is clear

 Literature Review

The literature review is adequate.

 Materials and Methods

 The second paragraph is confusing. It is not clear which part of the study is relevant.  Are the researchers referring to T1 or T2?

Table 1 – Schooling – does that refer to patients or parents?

Measures – when describing the measures, the modifiers such as powerful and valuable are not necessary.

 Results

This section is confusing and difficult to follow.   The results were expressed more clearly in the abstract. It is suggested that the authors follow the format of the abstract.

 Discussion:

It is suggested that the authors not include statistics in the Discussion section.  Rather, they should go into the Results Section.  The Discussion section would be enhanced with headings for the major points, i.e., traumatic stress, quality of life, etc.

Comments on the Quality of English Language

Round 2

Reviewer 1 Report

Comments and Suggestions for Authors

The authors have addressed the comments.

Comments on the Quality of English Language

Acceptable

Reviewer 2 Report

Comments and Suggestions for Authors

Good job revising the manuscript.  It is now much clearer.